# On Multiplicative Multitask Feature Learning

**Xin Wang**[†], **Jinbo Bi**[†], **Shipeng Yu**[‡], **Jiangwen Sun**[†]

[†]Dept. of Computer Science & Engineering
University of Connecticut
Storrs, CT 06269
wangxin,jinbo,javon@engr.uconn.edu

[‡] Health Services Innovation Center
Siemens Healthcare
Malvern, PA 19355
shipeng.yu@siemens.com

## Abstract

We investigate a general framework of multiplicative multitask feature learning which decomposes each task's model parameters into a multiplication of two components. One of the components is used across all tasks and the other component is task-specific. Several previous methods have been proposed as special cases of our framework. We study the theoretical properties of this framework when different regularization conditions are applied to the two decomposed components. We prove that this framework is mathematically equivalent to the widely used multitask feature learning methods that are based on a joint regularization of all model parameters, but with a more general form of regularizers. Further, an analytical formula is derived for the across-task component as related to the task-specific component for all these regularizers, leading to a better understanding of the shrinkage effect. Study of this framework motivates new multitask learning algorithms. We propose two new learning formulations by varying the parameters in the proposed framework. Empirical studies have revealed the relative advantages of the two new formulations by comparing with the state of the art, which provides instructive insights into the feature learning problem with multiple tasks.

## 1 Introduction

Multitask learning (MTL) captures and exploits the relationship among multiple related tasks and has been empirically and theoretically shown to be more effective than learning each task independently. Multitask feature learning (MTFL) investigates a basic assumption that different tasks may share a common representation in the feature space. Either the task parameters can be projected to explore the latent common substructure [18], or a shared low-dimensional representation of data can be formed by feature learning [10]. Recent methods either explore the latent basis that is used to develop the entire set of tasks, or learn how to group the tasks [16, 11], or identify if certain tasks are outliers to other tasks [6].

A widely used MTFL strategy is to impose a blockwise joint regularization of all task parameters to shrink the effects of features for the tasks. These methods employ a regularizer based on the so-called $\ell_{1,p}$ matrix norm [12, 13, 15, 22, 24] that is the sum of the $\ell_p$ norms of the rows in a matrix. Regularizers based on the $\ell_{1,p}$ norms encourage row sparsity. If rows represent features and columns represent tasks, they shrink the entire rows of the matrix to have zero entries. Typical choices for $p$ are 2 [15, 4] and $\infty$ [20] which are used in the very early MTFL methods. Effective algorithms have since then been developed for the $\ell_{1,2}$ [13] and $\ell_{1,\infty}$ [17] regularization. Later, the $\ell_{1,p}$ norm is generalized to include $1 < p \leq \infty$ with a probabilistic interpretation that the resultant MTFL method solves a relaxed optimization problem with a generalized normal prior for all tasks [22]. Recent research applies the capped $\ell_{1,1}$ norm as a nonconvex joint regularizer [5]. The major limitation of joint regularized MTFL is that it either selects a feature as relevant to all tasks or excludes it from all models, which is very restrictive in practice where tasks may share some features but may have their own specific features as well.

To overcome this limitation, one of the most effective strategies is to decompose the model parameters into either summation [9, 3, 6] or multiplication [21, 2, 14] of two components with different regularizers applied to the two components. One regularizer is used to take care of cross-task similarities and the other for cross-feature sparsity. Specifically, for the methods that decompose the parameter matrix into summation of two matrices, the dirty model in [9] employs $\ell_{1,1}$ and $\ell_{1,\infty}$ regularizers to the two components. A robust MTFL method in [3] uses the trace norm on one component for mining a low-rank structure shared by tasks and a column-wise $\ell_{1,2}$-norm on the other component for identifying task outliers. Another method applies the $\ell_{1,2}$-norm both row-wisely to one component and column-wisely to the other [6].

For the methods that work with multiplicative decompositions, the parameter vector of each task is decomposed into an element-wise product of two vectors where one is used across tasks and the other is task-specific. These methods either use the $\ell_2$-norm penalty on both of the component vectors [2], or the sparse $\ell_1$-norm on the two components (i.e., multi-level LASSO) [14]. The multi-level LASSO method has been analytically compared to the dirty model [14], showing that the multiplicative decomposition creates better shrinkage on the global and task-specific parameters. The across-task component can screen out the features irrelevant to all tasks. To exclude a feature from a task, the additive decomposition requires the corresponding entries in both components to be zero whereas the multiplicative decomposition only requires one of the components to have a zero entry. Although there are different ways to regularize the two components in the product, no systematic work has been done to analyze the algorithmic and statistical properties of the different regularizers. It is insightful to answer the questions such as how these learning formulations differ from the early methods based on blockwise joint regularization, how the optimal solutions of the two components look like, and how the resultant solutions are compared with those of other methods that also learn both shared and task-specific features.

In this paper, we investigate a general framework of the multiplicative decomposition that enables a variety of regularizers to be applied. This general form includes all early methods that represent model parameters as a product of two components [2, 14]. Our theoretical analysis has revealed that this family of methods is actually equivalent to the joint regularization based approach but with a more general form of regularizers, including those that do not correspond to a matrix norm. The optimal solution of the across-task component can be analytically computed by a formula of the optimal task-specific parameters, showing the different shrinkage effects. Statistical justification and efficient algorithms are derived for this family of formulations. Motivated by the analysis, we propose two new MTFL formulations. Unlike the existing methods [2, 14] where the same kind of vector norm is applied to both components, the shrinkage of the global and task-specific parameters differ in the new formulations. Hence, one component is regularized by the $\ell_2$-norm and the other is by the $\ell_1$-norm, which aims to reflect the degree of sparsity of the task-specific parameters relative to the sparsity of the across-task parameters. In empirical experiments, simulations have been designed to examine the various feature sharing patterns where a specific choice of regularizer may be preferred. Empirical results on benchmark data are also discussed.

## 2   The Proposed Multiplicative MTFL

Given $T$ tasks in total, for each task $t$, $t \in \{1, \cdots, T\}$, we have sample set $(\mathbf{X}_t \in \mathbb{R}^{\ell_t \times d}, \mathbf{y}_t \in \mathbb{R}^{\ell_t})$. The data set of $\mathbf{X}_t$ has $\ell_t$ examples, where the $i$-th row corresponds to the $i$-th example $\mathbf{x}_i^t$ of task $t$, $i \in \{1, \cdots, \ell_t\}$, and each column represents a feature. The vector $\mathbf{y}_t$ contains $y_i^t$, the label of the $i$-th example of task $t$. We consider functions of the linear form $\mathbf{X}_t \boldsymbol{\alpha}_t$ where $\boldsymbol{\alpha}_t \in \mathbb{R}^d$. We define the parameter matrix or weight matrix $\mathbf{A} = [\boldsymbol{\alpha}_1, \cdots, \boldsymbol{\alpha}_T]$ and $\boldsymbol{\alpha}^j$ are the rows, $j \in \{1, \cdots, d\}$.

A family of multiplicative MTFL methods can be derived by rewriting $\boldsymbol{\alpha}_t = \text{diag}(\mathbf{c})\boldsymbol{\beta}_t$ where $\text{diag}(\mathbf{c})$ is a diagonal matrix with its diagonal elements composing a vector $\mathbf{c}$. The $\mathbf{c}$ vector is used across all tasks, indicating if a feature is useful for any of the tasks, and the vector $\boldsymbol{\beta}_t$ is only for task $t$. Let $j$ index the entries in these vectors. We have $\alpha_j^t = c_j \beta_j^t$. Typically $\mathbf{c}$ comprises binary entries that are equal to 0 or 1, but the integer constraint is often relaxed to require just non-negativity. We minimize a regularized loss function as follows for the best $\mathbf{c}$ and $\boldsymbol{\beta}_{t:t=1,\cdots,T}$:

$$\min_{\boldsymbol{\beta}_t, \mathbf{c} \geq 0} \quad \sum_{t=1}^{T} L(\mathbf{c}, \boldsymbol{\beta}_t, \mathbf{X}_t, \mathbf{y}_t) + \gamma_1 \sum_{t=1}^{T} ||\boldsymbol{\beta}_t||_p^p + \gamma_2 ||\mathbf{c}||_k^k \qquad (1)$$

where $L(\cdot)$ is a loss function, e.g., the least squares loss for regression problems or the logistic loss for classification problems, $||\boldsymbol{\beta}_t||_p^p = \sum_{j=1}^d |\beta_j^t|^p$ and $||\mathbf{c}||_k^k = \sum_{j=1}^d (c_j)^k$, which are the $\ell_p$-norm of $\boldsymbol{\beta}_t$ to the power of $p$ and the $\ell_k$-norm of $\mathbf{c}$ to the power of $k$ if $p$ and $k$ are positive integers. The tuning parameters $\gamma_1$, $\gamma_2$ are used to balance the empirical loss and regularizers. At optimality, if $c_j = 0$, the $j$-th variable is removed for all tasks, and the corresponding row vector $\boldsymbol{\alpha}^j = \mathbf{0}$; otherwise the $j$-th variable is selected for use in at least one of the $\boldsymbol{\alpha}$'s. Then, a specific $\boldsymbol{\beta}_t$ can rule out the $j$-th variable from task $t$ if $\beta_j^t = 0$.

In particular, if both $p = k = 2$, Problem (1) becomes the formulation in [2] and if $p = k = 1$, Problem (1) becomes the formulation in [14]. Any other choices of $p$ and $k$ will derive into new formulations for MTFL. We first examine the theoretical properties of this entire family of methods, and then empirically study two new formulations by varying $p$ and $k$.

## 3 Theoretical Analysis

The joint $\ell_{1,p}$ regularized MTFL method minimizes $\sum_{t=1}^T L(\boldsymbol{\alpha}_t, \mathbf{X}_t, \mathbf{y}_t) + \lambda \sum_{j=1}^d ||\boldsymbol{\alpha}^j||_p$ for the best $\boldsymbol{\alpha}_{t:t=1,\cdots,T}$ where $\lambda$ is a tuning parameter. We now extend this formulation to allow more choices of regularizers. We introduce a new notation that is an operator applied to a vector, such as $\boldsymbol{\alpha}^j$. The operator $||\boldsymbol{\alpha}^j||^{p/q} = \sqrt[q]{\sum_{t=1}^T |\alpha_j^t|^p}$, $p, q \geq 0$, which corresponds to the $\ell_p$ norm if $p = q$ and both are positive integers. A joint regularized MTFL approach can solve the following optimization problem with pre-specified values of $p$, $q$ and $\lambda$, for the best parameters $\boldsymbol{\alpha}_{t:t=1,\cdots,T}$:

$$\min_{\boldsymbol{\alpha}_t} \quad \sum_{t=1}^T L(\boldsymbol{\alpha}_t, \mathbf{X}_t, \mathbf{y}_t) + \lambda \sum_{j=1}^d \sqrt{||\boldsymbol{\alpha}^j||^{p/q}}. \tag{2}$$

Our main result of this paper is (i) a theorem that establishes the equivalence between the models derived from solving Problem (1) and Problem (2) for properly chosen values of $\lambda$, $q$, $k$, $\gamma_1$ and $\gamma_2$; and (ii) an analytical solution of Problem (1) for $\mathbf{c}$ which shows how the sparsity of the across-task component is relative to the sparsity of task-specific components.

**Theorem 1** *Let $\hat{\boldsymbol{\alpha}}_t$ be the optimal solution to Problem (2) and $(\hat{\boldsymbol{\beta}}_t, \hat{\mathbf{c}})$ be the optimal solution to Problem (1). Then $\hat{\boldsymbol{\alpha}}_t = diag(\hat{\mathbf{c}})\hat{\boldsymbol{\beta}}_t$ when $\lambda = 2\sqrt{\gamma_1^{2-\frac{p}{kq}}\gamma_2^{\frac{p}{kq}}}$ and $q = \frac{k+p}{2k}$ (or $k = \frac{p}{2q-1}$).*

 **Proof.** The theorem holds by proving the following two Lemmas. The first lemma proves that the solution $\hat{\boldsymbol{\alpha}}_t$ of Problem (2) also minimizes the following optimization problem:

$$\min_{\boldsymbol{\alpha}_t, \boldsymbol{\sigma} \geq 0} \quad \sum_{t=1}^T L(\boldsymbol{\alpha}_t, \mathbf{X}_t, \mathbf{y}_t) + \mu_1 \sum_{j=1}^d \sigma_j^{-1} ||\boldsymbol{\alpha}^j||^{p/q} + \mu_2 \sum_{j=1}^d \sigma_j, \tag{3}$$

and the optimal solution of Problem (3) also minimizes Problem (2) when proper values of $\lambda$, $\mu_1$ and $\mu_2$ are chosen. The second lemma connects Problem (3) to our formulation (1). We show that the optimal $\hat{\sigma}_j$ is equal to $(\hat{c}_j)^k$, and then the optimal $\hat{\boldsymbol{\beta}}$ can be computed from the optimal $\hat{\boldsymbol{\alpha}}$. ∎

**Lemma 1** *The solution sets of Problem (2) and Problem (3) are identical when $\lambda = 2\sqrt{\mu_1\mu_2}$.*

 **Proof.** First, we show that when $\lambda = 2\sqrt{\mu_1\mu_2}$, the optimal solution $\hat{\alpha}_j^t$ of Problem (2) minimizes Problem (3) and the optimal $\hat{\sigma}_j = \mu_1^{\frac{1}{2}}\mu_2^{-\frac{1}{2}}\sqrt{||\hat{\boldsymbol{\alpha}}^j||^{p/q}}$. By the Cauchy-Schwarz inequality, the following inequality holds

$$\mu_1 \sum_{j=1}^d \sigma_j^{-1} ||\boldsymbol{\alpha}^j||^{p/q} + \mu_2 \sum_{j=1}^d \sigma_j \geq 2\sqrt{\mu_1\mu_2} \sum_{j=1}^d \sqrt{||\boldsymbol{\alpha}^j||^{p/q}}$$

where the equality holds if and only if $\sigma_j = \mu_1^{\frac{1}{2}}\mu_2^{-\frac{1}{2}}\sqrt{||\boldsymbol{\alpha}^j||^{p/q}}$. Since Problems (3) and (2) use the exactly same loss function, when we set $\hat{\sigma}_j = \mu_1^{\frac{1}{2}}\mu_2^{-\frac{1}{2}}\sqrt{||\hat{\boldsymbol{\alpha}}^j||^{p/q}}$, Problems (3) and (2) have identical objective function if $\lambda = 2\sqrt{\mu_1\mu_2}$. Hence the pair $(\hat{\mathbf{A}} = (\hat{\alpha}_j^t)_{jt}, \hat{\boldsymbol{\sigma}} = (\hat{\sigma}_j)_{j=1,\cdots,d})$ minimizes Problem (3) as it entails the objective function to reach its lower bound.

Second, it can be proved that if the pair $(\hat{\mathbf{A}}, \hat{\boldsymbol{\sigma}})$ minimizes Problem (3), then $\hat{\mathbf{A}}$ also minimizes Problem (2) by proof of contradiction. Suppose that $\hat{\mathbf{A}}$ does not minimize Problem (2), which means that there exists $\tilde{\boldsymbol{\alpha}}^j$ ($\neq \hat{\boldsymbol{\alpha}}^j$ for some $j$) that is an optimal solution to Problem (2) and achieves a lower objective value than $\hat{\boldsymbol{\alpha}}^j$. We set $\tilde{\sigma}_j = \mu_1^{\frac{1}{2}} \mu_2^{-\frac{1}{2}} \sqrt{||\tilde{\boldsymbol{\alpha}}^j||^{p/q}}$. The pair $(\tilde{\mathbf{A}}, \tilde{\boldsymbol{\sigma}})$ is an optimal solution of Problem (3) as proved in the first paragraph. Then $(\tilde{\mathbf{A}}, \tilde{\boldsymbol{\sigma}})$ will bring the objective function of Problem (3) to a lower value than that of $(\hat{\mathbf{A}}, \hat{\boldsymbol{\sigma}})$, contradicting to the assumption that $(\hat{\mathbf{A}}, \hat{\boldsymbol{\sigma}})$ be optimal to Problem (3).

Hence, we have proved that Problems (3) and (2) have identical solutions when $\lambda = 2\sqrt{\mu_1 \mu_2}$. ∎

From the proof of Lemma (1), we also see that the optimal objective value of Problem (2) gives a lower bound to the objective of Problem (3). Let $\sigma_j = (c_j)^k$, $k \in \mathbb{R}$, $k \neq 0$ and $\alpha_j^t = c_j \beta_j^t$, an equivalent objective function of Problem (3) can be derived.

**Lemma 2** *The optimal solution $(\hat{\mathbf{A}}, \hat{\boldsymbol{\sigma}})$ of Problem (3) is equivalent to the optimal solution $(\hat{\mathbf{B}}, \hat{\mathbf{c}})$ of Problem (1) where $\hat{\alpha}_j^t = \hat{c}_j \hat{\beta}_j^t$ and $\hat{\sigma}_j = (\hat{c}_j)^k$ when $\gamma_1 = \mu_1^{\frac{kq}{2kq-p}} \mu_2^{\frac{kq-p}{2kq-p}}$, $\gamma_2 = \mu_2$, and $k = \frac{p}{2q-1}$.*
**Proof.** First, by proof of contradiction, we show that if $\hat{\alpha}_j^t$ and $\hat{\sigma}_j$ optimize Problem (3), then $\hat{c}_j = \sqrt[k]{\hat{\sigma}_j}$ and $\hat{\beta}_j^t = \frac{\hat{\alpha}_j^t}{\hat{c}_j}$ optimize Problem (1). Denote the objectives of (1) and (3) by $J^{(1)}$ and $J^{(3)}$. Substituting $\hat{\beta}_j^t$, $\hat{c}_j$ for $\hat{\alpha}_j^t$, $\hat{\sigma}_j$ in $J^{(3)}$ yields an objective function $L(\hat{\mathbf{c}}, \hat{\boldsymbol{\beta}}_t, \mathbf{X}_t, \mathbf{y}_t) + \mu_1 \sum_{j=1}^d ||\hat{\boldsymbol{\beta}}^j||^{p/q} \hat{c}_j^{(p-kq)/q} + \mu_2 \sum_{j=1}^d (\hat{c}_j)^k$. By the proof of Lemma 1, $\hat{\sigma}_j = \mu_1^{\frac{1}{2}} \mu_2^{-\frac{1}{2}} \sqrt{||\hat{\boldsymbol{\alpha}}^j||^{p/q}}$. Hence, $\hat{c}_j = \left(\mu_1 \mu_2^{-1} ||\hat{\boldsymbol{\beta}}^j||^{p/q}\right)^{\frac{q}{2kq-p}}$. Applying the formula of $\hat{c}_j$ and substituting $\mu_1$ and $\mu_2$ by $\gamma_1$ and $\gamma_2$ yield an objective identical to $J^{(1)}$. Suppose $\exists (\tilde{\beta}_j^t, \tilde{c}_j)(\neq (\hat{\beta}_j^t, \hat{c}_j))$ that minimize (1), and $J^{(1)}(\tilde{\beta}_j^t, \tilde{c}_j) < J^{(1)}(\hat{\beta}_j^t, \hat{c}_j)$. Let $\tilde{\alpha}_j^t = \tilde{c}_j \tilde{\beta}_j^t$ and substitute $\tilde{\beta}_j^t$ by $\tilde{\alpha}_j^t / \tilde{c}_j$ in $J^{(1)}$. By Cauchy-Schwarz inequality, we similarly have $\tilde{c}_j = (\gamma_1 \gamma_2^{-1} \sum_{t=1}^T (\tilde{\alpha}_j^t)^p)^{\frac{1}{p+k}}$. Thus, $J^{(1)}(\tilde{\alpha}_j^t, \tilde{c}_j)$ can be derived into $J^{(3)}(\tilde{\alpha}_j^t, \tilde{c}_j)$. Let $\tilde{\sigma}_j = (\tilde{c}_j)^k$, and we have $J^{(3)}(\tilde{\alpha}_j^t, \tilde{\sigma}_j) < J^{(3)}(\hat{\alpha}_j^t, \hat{\sigma}_j)$, which contradicts with the optimality of $(\hat{\alpha}_j^t, \hat{\sigma}_j)$. Second, we similarly prove that if $\hat{\beta}_j^t$ and $\hat{c}_j$ optimize Problem (1), then $\hat{\alpha}_j^t = \hat{c}_j \hat{\beta}_j^t$ and $\hat{\sigma}_j = (\hat{c}_j)^k$ optimize Problem (3). ∎

Now, combining the results from the two Lemmas, we can derive that when $\lambda = 2\sqrt{\gamma_1^{2-\frac{p}{kq}} \gamma_2^{\frac{p}{kq}}}$ and $q = \frac{k+p}{2k}$, the optimal solutions to Problems (1) and (2) are equivalent. Solving Problem (1) will yield an optimal solution $\hat{\boldsymbol{\alpha}}$ to Problem (2) and vice versa.

**Theorem 2** *Let $\hat{\boldsymbol{\beta}}_t$, $t = 1, \cdots, T$, be the optimal solutions of Problem (1), Let $\hat{\mathbf{B}} = [\hat{\boldsymbol{\beta}}_1, \cdots, \hat{\boldsymbol{\beta}}_T]$ and $\hat{\boldsymbol{\beta}}^j$ denote the $j$-th row of the matrix $\hat{\mathbf{B}}$. Then,*

$$\hat{c}_j = (\gamma_1/\gamma_2)^{\frac{1}{k}} ||\hat{\boldsymbol{\beta}}^j||^{\frac{p}{2kq-p}}, \tag{4}$$

*for all $j = 1, \cdots, d$, is optimal to Problem (1).*

**Proof.** This analytical formula can be directly derived from Lemma 1 and Lemma 2. When we set $\hat{\sigma}_j = (\hat{c}_j)^k$ and $\hat{\alpha}_j^t = \hat{c}_j \hat{\beta}_j^t$ in Problem (3), we obtain $\hat{c}_j = \left(\mu_1 \mu_2^{-1} ||\hat{\boldsymbol{\beta}}^j||^{p/q}\right)^{\frac{q}{2kq-p}}$. In the proof of Lemma 2, we obtain that $\mu_1 = \gamma_1^{\frac{2kq-p}{kq}} \gamma_2^{\frac{p-kq}{kq}}$ and $\mu_2 = \gamma_2$. Substituting these formula into the formula of $\mathbf{c}$ yields the formula (4). ∎

Based on the derivation, for each pair of $\{p, q\}$ and $\lambda$ in Problem (2), there exists an equivalent problem (1) with determined values of $k$, $\gamma_1$ and $\gamma_2$, and vice versa. Note that if $q = p/2$, the regularization term on $\boldsymbol{\alpha}^j$ in Problem (2) becomes the standard $p$-norm. In particular, if $\{p, q\} = \{2, 1\}$ in Problem (2) as used in the methods of [15] and [1], the $\ell_2$-norm regularizer is applied to $\boldsymbol{\alpha}^j$. Then, this problem is equivalent to Problem (1) when $k = 2$ and $\lambda = 2\sqrt{\gamma_1 \gamma_2}$, the same formulation in [2]. If $\{p, q\} = \{1, 1\}$, the square root of $\ell_1$-norm regularizer is applied to $\boldsymbol{\alpha}^j$. Our theorem 1 shows that this problem is equivalent to the multi-level LASSO MTFL formulation [14] which is Problem (1) with $k = 1$ and $\lambda = 2\sqrt{\gamma_1 \gamma_2}$.

# 4 Probabilistic Interpretation

In this section we show the proposed multiplicative formalism is related to the *maximum a posteriori* (MAP) solution of a probabilistic model. Let $p(\mathbf{A}|\boldsymbol{\Delta})$ be the prior distribution of the weight matrix $\mathbf{A} = [\boldsymbol{\alpha}_1, \ldots, \boldsymbol{\alpha}_T] = [\boldsymbol{\alpha}^{1\top}, \ldots, \boldsymbol{\alpha}^{d\top}]^\top \in \mathbb{R}^{d \times T}$, where $\boldsymbol{\Delta}$ denote the parameter of the prior. Then the *a posteriori* distribution of $\mathbf{A}$ can be calculated via Bayes rule as $p(\mathbf{A}|\mathbf{X}, \mathbf{y}, \boldsymbol{\Delta}) \propto p(\mathbf{A}|\boldsymbol{\Delta}) \prod_{t=1}^{T} p(\mathbf{y}_t|\mathbf{X}_t, \boldsymbol{\alpha}_t)$. Denote $z \sim \mathcal{GN}(\mu, \rho, q)$ the *univariate generalized normal distribution*, with the density function $p(z) = \frac{1}{2\rho\Gamma(1+1/q)} \exp(-\frac{|z-\mu|^q}{\rho^q})$, in which $\rho > 0$, $q > 0$, and $\Gamma(\cdot)$ is the Gamma function [7]. Now let each element of $\mathbf{A}$, $\alpha_j^t$, follow a generalized normal prior, $\alpha_j^t \sim \mathcal{GN}(0, \delta_j, q)$. Then with the i.i.d. assumption, the prior takes the form (also refer to [22] for a similar treatment)

$$p(\mathbf{A}|\boldsymbol{\Delta}) \propto \prod_{j=1}^{d} \prod_{t=1}^{T} \frac{1}{\delta_j} \exp\left(-\frac{|\alpha_j^t|^q}{\delta_j^q}\right) = \prod_{j=1}^{d} \frac{1}{\delta_j^T} \exp\left(-\frac{\|\boldsymbol{\alpha}^j\|_q^q}{\delta_j^q}\right), \tag{5}$$

where $\|\cdot\|_q$ denote vector $q$-norm. With an appropriately chosen likelihood function $p(\mathbf{y}_t|\mathbf{X}_t, \boldsymbol{\alpha}_t) \propto \exp(-L(\boldsymbol{\alpha}_t, \mathbf{X}_t, \mathbf{y}_t))$, finding the MAP solution is equivalent to solving the following problem: $\min_{\mathbf{A}, \boldsymbol{\Delta}} J = \sum_{t=1}^{T} L(\boldsymbol{\alpha}_t, \mathbf{X}_t, \mathbf{y}_t) + \sum_{j=1}^{d} \left(\frac{\|\boldsymbol{\alpha}^j\|_q^q}{\delta_j^q} + T \ln \delta_j\right)$. By setting the derivative of $J$ with respect to $\delta_j$ to zero, we obtain:

$$\min_{\mathbf{A}} \quad J = \sum_{t=1}^{T} L(\boldsymbol{\alpha}_t, \mathbf{X}_t, \mathbf{y}_t) + T \sum_{j=1}^{d} \ln \|\boldsymbol{\alpha}^j\|_q. \tag{6}$$

Now let us look at the multiplicative nature of $\alpha_j^t$ with different $q \in [1, \infty]$. When $q = 1$, we have:

$$\sum_{j=1}^{d} \ln \|\boldsymbol{\alpha}^j\|_1 = \sum_{j=1}^{d} \ln \left(\sum_{t=1}^{T} |\alpha_j^t|\right) = \sum_{j=1}^{d} \ln \left(\sum_{t=1}^{T} |c_j \beta_j^t|\right) = \sum_{j=1}^{d} \left(\ln |c_j| + \ln \sum_{t=1}^{T} |\beta_j^t|\right). \tag{7}$$

Because of $\ln z \le z - 1$ for any $z > 0$, we can optimize an upper bound of $J$ in (6). We then have $\min_{\mathbf{A}} J_1 = \sum_{t=1}^{T} L(\boldsymbol{\alpha}_t, \mathbf{X}_t, \mathbf{y}_t) + T \sum_{j=1}^{d} |c_j| + T \sum_{j=1}^{d} \sum_{t=1}^{T} |\beta_j^t|$, which is equivalent to the multiplicative formulation (1) where $\{p, k\} = \{1, 1\}$. For $q > 1$, we have:

$$\sum_{j=1}^{d} \ln \|\boldsymbol{\alpha}^j\|_q = \frac{1}{q} \sum_{j=1}^{d} \ln \left(\sum_{t=1}^{T} |c_j \beta_j^t|^q\right) \le \frac{1}{q} \sum_{j=1}^{d} \ln \left(\max\{|c_1|, \ldots, |c_d|\}^q \cdot \sum_{t=1}^{T} |\beta_j^t|^q\right) \tag{8}$$

$$= \sum_{j=1}^{d} \ln \|\mathbf{c}\|_\infty + \frac{1}{q} \sum_{j=1}^{d} \ln \sum_{t=1}^{T} |\beta_j^t|^q \le d\|\mathbf{c}\|_\infty + \frac{1}{q} \sum_{t=1}^{T} \|\boldsymbol{\beta}_t\|_q^q - (d + \frac{d}{q}). \tag{9}$$

Since vector norms satisfy $\|\mathbf{z}\|_\infty \le \|\mathbf{z}\|_k$ for any vector $\mathbf{z}$ and $k \ge 1$, these inequalities lead to an upper bound of $J$ in (6), i.e., $\min_{\mathbf{A}} \quad J_{q,k} = \sum_{t=1}^{T} L(\boldsymbol{\alpha}_t, \mathbf{X}_t, \mathbf{y}_t) + Td\|\mathbf{c}\|_k + \frac{T}{q} \sum_{t=1}^{T} \|\boldsymbol{\beta}_t\|_q^q$, which is equivalent to the general multiplicative formulation in (1).

# 5 Optimization Algorithm

Alternating optimization algorithms have been used in both of the early methods [2, 14] to solve Problem (1) which alternate between solving two subproblems: solve for $\boldsymbol{\beta}_t$ with fixed $\mathbf{c}$; solve for $\mathbf{c}$ with fixed $\boldsymbol{\beta}_t$. The convergence property of such an alternating algorithm has been analyzed in [2] that it converges to a local minimizer. However, both subproblems in the existing methods can only be solved using iterative algorithms such as gradient descent, linear or quadratic program solvers. We design a new alternating optimization algorithm that utilizes the property that both Problems (1) and (2) are equivalent to Problem (3) used in our proof and we derive a closed-form solution for $\mathbf{c}$ for the second subproblem. The following theorem characterizes this result.

**Theorem 3** *For any given values of $\boldsymbol{\alpha}_{t:t=1,\cdots,T}$, the optimal $\boldsymbol{\sigma}$ of Problem (3) when $\boldsymbol{\alpha}_{t:t=1,\cdots,T}$ are fixed to the given values can be computed by $\sigma_j = \gamma_1^{1-\frac{p}{2kq}} \gamma_2^{\frac{1}{2}-\frac{p}{2kp}} \sqrt[2q]{\sum_{t=1}^{T} (\alpha_j^t)^p}$, $j = 1, \cdots, d$.*

**Proof.** By the Cauchy-Schwarz inequality and the same argument used in the proof of Lemma 1, we obtain that the best $\sigma$ for a given set of $\alpha_{t:t=1,\cdots,T}$ is $\sigma_j = \mu_1^{\frac{1}{2}}\mu_2^{-\frac{1}{2}}\sqrt{||\alpha^j||^{p/q}}$. We also know that $\mu_1$ and $\mu_2$ are chosen in such a way that $\gamma_1 = \mu_1^{\frac{kq}{2kq-p}}\mu_2^{\frac{kq-p}{2kq-p}}$ and $\gamma_2 = \mu_2$. This is equivalent to have $\mu_1 = \gamma_1^{\frac{2kq-p}{kq}}\gamma_2^{\frac{p-kq}{kq}}$ and $\mu_2 = \gamma_2$. Substituting them into the formula of $\sigma$ yields the result. ∎

Now, in the algorithm to solve Problem (1), we solve the first subproblem to obtain a new iterate $\beta_t^{new}$, then we use the current value of $\mathbf{c}$, $\mathbf{c}^{old}$, to compute the value of $\alpha_t^{new} = \mathrm{diag}(\mathbf{c}^{old})\beta_t^{new}$, which is then used to compute $\sigma_j$ according to the formula in Theorem 3. Then, $\mathbf{c}$ is computed as $c_j = \sqrt[k]{\sigma_j}$, $j = 1, \cdots, d$. The overall procedure is summarized in Algorithm 1.

---

**Algorithm 1** Alternating optimization for multiplicative MTFL

> **Input:** $\mathbf{X}_t, \mathbf{y}_t, t = 1, \cdots, T$, as well as $\gamma_1, \gamma_2, p$ and $k$
> **Initialize:** $c_j = 1, \forall j = 1, \cdots, d$
> **repeat**
>     **1.** Convert $\mathbf{X}_t \mathrm{diag}(\mathbf{c}^{s-1}) \rightarrow \tilde{\mathbf{X}}_t, \forall\, t = 1, \cdots, T$
>     **for** $t = 1, \cdots, T$ **do**
>         Solve $\min_{\beta_t} L(\beta_t, \tilde{\mathbf{X}}_t, \mathbf{y}_t) + \gamma_1 ||\beta_t||_p^p$ for $\beta_t^s$
>     **end for**
>     **2.** Compute $\alpha_t^s = \mathrm{diag}(\mathbf{c}^{(s-1)})\beta_t^s$, and compute $\mathbf{c}^s$ as $c_j^s = \sqrt[k]{\sigma_j}$ where $\sigma_j$ is computed according to the formula in Theorem 3.
> **until** $\max(|(\alpha_j^t)^s - (\alpha_j^t)^{s-1}|) < \epsilon$
> **Output:** $\alpha_t, \mathbf{c}$ and $\beta_t, t = 1, \cdots, T$

---

Algorithm 1 can be used to solve the entire family of methods characterized by Problem (1). The first subproblem involves convex optimization if a convex loss function is chosen and $p \geq 1$, and can be solved separately for individual tasks using single task learning. The second subproblem is analytically solved by a formula that guarantees that Problem (1) reaches a lower bound for the current $\alpha_t$. In this paper, the least squares and logistic regression losses are used and both of them are convex and differentiable loss functions. When convex and differentiable losses are used, theoretical results in [19] can be used to prove the convergence of the proposed algorithm. We choose to monitor the maximum norm of the $\mathbf{A}$ matrix to terminate the process, but it can be replaced by any other suitable termination criterion. Initialization can be important for this algorithm, and we suggest starting with $\mathbf{c} = \mathbf{1}$, which considers all features initially in the learning process.

## 6 Two New Formulations

The two existing methods discussed in [2, 14] use $p = k$ in their formulations, which renders $\beta_j^t$ and $c_j$ the same amount of shrinkage. To explore other feature sharing patterns among tasks, we propose two new formulations where $p \neq k$. For the common choices of $p$ and $k$, the relation between the optimal $\mathbf{c}$ and $\beta$ can be computed according to Theorem 2, and is summarized in Table 1.

1. When the majority of the features is not relevant to any of the tasks, it requires a sparsity-inducing norm on $\mathbf{c}$. However, within the relevant features, many features are shared between tasks. In other words, the features used in each task are not sparse relative to all the features selected by $\mathbf{c}$, which requires a non-sparsity-inducing norm on $\beta$. Hence, we use $\ell_1$ norm on $\mathbf{c}$ and $\ell_2$ norm on all $\beta$'s in Formulation (1). This formulation is equivalent to the joint regularization method of $\min_{\alpha_t} \sum_{t=1}^T L(\alpha_t, \mathbf{X}_t, \mathbf{y}_t) + \lambda \sum_{j=1}^d \sqrt[3]{\sum_{t=1}^T (\alpha_j^t)^2}$ where $\lambda = 2\gamma_1^{\frac{1}{3}}\gamma_2^{\frac{2}{3}}$.

2. When many or all features are relevant to the given tasks, it may prefer the $\ell_2$ norm penalty on $\mathbf{c}$. However, only a limited number of features are shared between tasks, i.e., the features used by individual tasks are sparse with respect to the features selected as useful across tasks by $\mathbf{c}$. We can impose the $\ell_1$ norm penalty on $\beta$. This formulation is equivalent to the joint regularization method of $\min_{\alpha_t} \sum_{t=1}^T L(\alpha_t, \mathbf{X}_t, \mathbf{y}_t) + \lambda \sum_{j=1}^d \sqrt[3]{\left(\sum_{t=1}^T |\alpha_j^t|\right)^2}$ where $\lambda = 2\gamma_1^{\frac{2}{3}}\gamma_2^{\frac{1}{3}}$.

Table 1: The shrinkage effect of $\mathbf{c}$ with respect to $\boldsymbol{\beta}$ for four common choices of $p$ and $k$.

| $p$ | $k$ | $\mathbf{c}$ | $p$ | $k$ | $\mathbf{c}$ |
|---|---|---|---|---|---|
| 2 | 2 | $\hat{c}_j = \sqrt{\gamma_1 \gamma_2^{-1}} \sqrt{\sum_{t=1}^T \hat{\beta}_j^{t\,2}}$ | 2 | 1 | $\hat{c}_j = \gamma_1 \gamma_2^{-1} \sum_{t=1}^T \hat{\beta}_j^{t\,2}$ |
| 1 | 1 | $\hat{c}_j = \gamma_1 \gamma_2^{-1} \sum_{t=1}^T |\hat{\beta}_j^t|$ | 1 | 2 | $\hat{c}_j = \sqrt{\gamma_1 \gamma_2^{-1}} \sqrt{\sum_{t=1}^T |\hat{\beta}_j^t|}$ |

## 7 Experiments

In this section, we empirically evaluate the performance of the proposed multiplicative MTFL with the four parameter settings listed in Table 1 on synthetic and real-world data for both classification and regression problems. The first two settings $(p, k) = (2, 2), (1, 1)$ give the same methods respectively in [2, 14], and the last two settings correspond to our new formulations. The least squares and logistic regression losses are used, respectively, for regression and classification problems. We focus on the understanding of the shrinkage effects created by the different choices of regularizers in multiplicative MTFL. These methods are referred to as MMTFL and are compared with the dirty model (DMTL) [9] and robust MTFL (rMTFL) [6] that use the additive decomposition.

The first subproblem of Algorithm 1 was solved using CPLEX solvers and single task learning in the initial first subproblem served as baseline. We used respectively 25%, 33% and 50% of the available data in each data set for training and the rest data for test. We repeated the random split 15 times and reported the averaged performance. For each split, the regularization parameters of each method were tuned by a 3-fold cross validation within the training data. The regression performance was measured by the coefficient of determination, denoted as $R^2$, which was computed as 1 minus the ratio of the sum of squared residuals and the total sum of squares. The classification performance was measured by the F1 score, which was the harmonic mean of precision and recall.

**Synthetic Data.** We created two synthetic data sets which included 10 and 20 tasks, respectively. For each task, we created 200 examples using 100 features with pre-defined combination weights $\boldsymbol{\alpha}$. Each feature was generated following the $N(\mathbf{0}, \mathbf{1})$ distribution. We added noise and computed $\mathbf{y}_t = \mathbf{X}_t \boldsymbol{\alpha}_t + \boldsymbol{\epsilon}_t$ for each task $t$ where the noise $\epsilon$ followed a distribution $N(0, 1)$. We put the different tasks' $\boldsymbol{\alpha}$'s together as rows in Figure 1. The values of $\boldsymbol{\alpha}$'s were specified in such a way for us to explore how the structure of feature sharing influences the multitask learning models when various regularizers are used. In particular, we illustrate the cases where the two newly proposed formulations outperformed other methods.

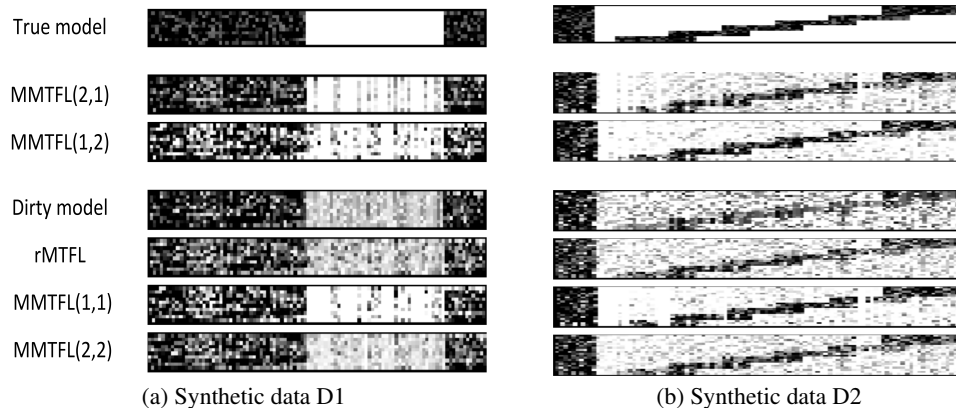

(a) Synthetic data D1          (b) Synthetic data D2

Figure 1: Parameter matrix learned by different methods (darker color indicates greater values.).

*Synthetic Data 1 (D1).* As shown in Figure 1a, 40% of components in all $\boldsymbol{\alpha}$'s were set to 0, and these features were irrelevant to all tasks. The rest features were used in every task's model and hence these models were sparse with respect to all of the features, but not sparse with respect to the selected features. This was the assumption for the early joint regularized methods to work. To learn this feature sharing structure, however, we observed that the amount of shrinkage needed would be different for $\mathbf{c}$ and $\boldsymbol{\beta}$. This case might be in favor of the $\ell_1$ norm penalty on $\mathbf{c}$.

*Synthetic Data 2 (D2).* The designed parameter matrix is shown in Figure 1b where tasks were split into 6 groups. Five features were irrelevant to all tasks, 10 features were used by all tasks, and each

of the remaining 85 features was used by only 1 or 2 groups. The neighboring groups of tasks in Figure 1b shared only 7 features besides those 10 common features. Non-neighboring tasks did not share additional features. We expected **c** to be non-sparse. However, each task only used very few features with respect to all available features, and hence each $\beta$ should be sparse.

Figure 1 shows the parameter matrices (with columns representing features for illustrative convenience) learned by different methods using 33% of the available examples in each data set. We can clearly see that MMTFL(2,1) performs the best for Synthetic data D1. This result suggests that the classic choices of using $\ell_2$ or $\ell_1$ penalty on both **c** and $\beta$ (corresponding to early joint regularized methods) might not always be optimal. MMTFL(1,2) is superior for Synthetic data D2, where each model shows strong feature sparsity but few features can be removed if all tasks are considered. Table 2 summarizes the performance comparison where the best performance is highlighted in bold font. Note that the feature sharing patterns may not be revealed by the recent methods on clustered multitask learning that cluster tasks into groups [10, 8, 23] because no cluster structure is present in Figure 1b, for instance. Rather, the sharing pattern in Figure 1b is in the shape of staircase.

Table 2: Comparison of the performance between various multitask learning models

| Data set | | STL | DMTL | rMTFL | MMTFL(2,2) | MMTFL(1,1) | MMTFL(2,1) | MMTFL(1,2) |
|---|---|---|---|---|---|---|---|---|
| *Synthetic data* | | | | | | | | |
| **D1** ($R^2$) | 25% | 0.40±0.02 | 0.60±0.02 | 0.58±0.02 | 0.64±0.02 | 0.54±0.03 | **0.73±0.02** | 0.42±0.04 |
| | 33% | 0.55±0.03 | 0.73±0.01 | 0.61±0.02 | 0.79±0.02 | 0.76±0.01 | **0.86±0.01** | 0.65±0.03 |
| | 50% | 0.60±0.02 | 0.75±0.01 | 0.66±0.01 | 0.86±0.01 | 0.88±0.01 | **0.90±0.01** | 0.84±0.01 |
| **D2** ($R^2$) | 25% | 0.28±0.02 | 0.36±0.01 | 0.46±0.01 | 0.45±0.01 | 0.35±0.05 | 0.46±0.02 | **0.49±0.02** |
| | 33% | 0.35±0.01 | 0.42±0.02 | 0.63±0.03 | 0.69±0.02 | 0.75±0.01 | 0.67±0.03 | **0.83±0.02** |
| | 50% | 0.75±0.01 | 0.81±0.01 | 0.83±0.01 | 0.91±0 | 0.95±0 | 0.92±0.01 | **0.97±0** |
| *Real-world data* | | | | | | | | |
| SARCOS | 25% | 0.78±0.02 | **0.90± 0** | **0.90±0** | 0.89± 0 | 0.89± 0 | **0.90±0.01** | 0.87±0.01 |
| ($R^2$) | 33% | 0.78±0.02 | 0.88±0.11 | 0.89±0.1 | 0.90± 0 | 0.90± 0 | **0.91±0.01** | 0.89±0.01 |
| | 50% | 0.83±0.06 | 0.87± 0.1 | 0.89±0.1 | **0.91± 0** | 0.90± 0.01 | **0.91±0.01** | 0.89±0.01 |
| USPS | 25% | 0.83±0.01 | 0.89±0.01 | **0.91±0.01** | 0.90±0.01 | 0.90±0.01 | 0.90±0.01 | **0.91±0.01** |
| (F1 score) | 33% | 0.84±0.02 | 0.90±0.01 | 0.90±0.01 | 0.89±0.01 | 0.90±0.01 | 0.90±0.01 | **0.91±0.01** |
| | 50% | 0.87±0.02 | 0.91±0.01 | 0.92±0.01 | 0.92±0.01 | 0.92±0.01 | 0.92±0.01 | **0.93±0.01** |

**Real-world Data.** Two benchmark data sets, the Sarcos [1] and the USPS data sets [10], were used for regression and classification tests respectively. The Sarcos data set has 48,933 observations and each observation (example) has 21 features. Each task is to map from the 21 features to one of the 7 consecutive torques of the Sarcos robot arm. We randomly selected 2000 examples for use in each task. USPS handwritten digits data set has 2000 examples and 10 classes as the digits from 0 to 9. We first used principle component analysis to reduce the feature dimension to 87. To create binary classification tasks, we randomly chose images from the other 9 classes to be the negative examples. Table 2 provides the performance of the different methods on these two data sets, which shows the effectiveness of MMTFL(2,1) and MMTFL(1,2).

## 8  Conclusion

In this paper, we study a general framework of multiplicative multitask feature learning. By decomposing the model parameter of each task into a product of two components: the across-task feature indicator and task-specific parameters, and applying different regularizers to the two components, we can select features for individual tasks and also search for the shared features among tasks. We have studied the theoretical properties of this framework when different regularizers are applied and found that this family of methods creates models equivalent to those of the joint regularized MTL methods but with a more general form of regularization. Further, an analytical formula is derived for the across-task component as related to the task-specific component, which shed light on the different shrinkage effects in the various regularizers. An efficient algorithm is derived to solve the entire family of methods and also tested in our experiments. Empirical results on synthetic data clearly show that there may not be a particular choice of regularizers that is universally better than other choices. We empirically show a few feature sharing patterns that are in favor of two newly-proposed choices of regularizers, which is confirmed on both synthetic and real-world data sets.

### Acknowledgements
Jinbo Bi and her students Xin Wang and Jiangwen Sun were supported by NSF grants IIS-1320586, DBI-1356655, IIS-1407205, and IIS-1447711.

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
