[Reviews · NeurIPS 2014]

Submitted by Assigned_Reviewer_5

Thanks to the authors for their clarifying rebuttal.

This paper connects two previously described methods of multitask feature learning, one where regularization is applied within and across tasks separately, and one where the regularization is performed on the jointly learned parameter matrix. This paper proves that under specific parameter settings, the two formulations are equivalent.

This paper is fairly clear, but I would have liked one final statement of the full form of problem 1 using problem 2's parameters (or vice versa) rather than the current stating of parameter equivalence. However, the proofs are nicely presented with sketches of the steps described beforehand to help the reader stitch together the parts.

I'm unclear on the evaluation described in section 7. Is 3-fold cross validation performed on a 25,33,50% subset of the data? Is that selection of data to use for cross validation itself selected at random 15 times, or is the partitioning of the subset done randomly 15 times? How different are the chosen regularization parameters across the 15 cross validations?

I have only minor quibbles about the results - for both synthetic datasets, I think it's hard to determine visually that any method performs better than MMTFL(1,1). It's obvious from the results in Table 2 that MMTFL(1,1) is not superior, but eyeballing this weight matrix does not make that clear.

Minor comments
I've never heard of equations referred to as "problems" before, rather they're usually referred to as "equation 1" and "equation 2".
Table 2 should have a heading for the x-val percentage column.
On line 312 "features is" -> "features are"
and line 320 "features is" -> "features are" ( since the authors made this word choice twice perhaps they think the verb refers to a singular "majority" or "limited number", which I think of as plural)
Summary: This paper connects two formulations of multitask feature learning by proving their equivalence under certain circumstances. This reformulation leads to two new parameter settings which is found to perform quite well on real-world data.

Submitted by Assigned_Reviewer_17

The paper proposes a general multi-task learning framework, in which the model parameter is decomposed into a multiplication of two components. The paper conducts theoretical analysis on the proposed framework and also discusses its optimization algorithms.

The most interesting point of this work seems to be the close connections to the existing formulations in [2] and [14]. These connections are obvious after the equivalence of Eq (1) and Eq (2) is setup.

A possible limitation of the proposed method is its non-convexity. The non-convexity makes the model selection difficult - it is difficult to know which regularization parameter setting is better when the obtained solution is always local.

From Algorithm 1 (the alternating optimization algorithm), the computational cost may be high, when the task number is large.
Summary: The paper proposes a general multi-task learning framework, in which the model parameter is decomposed into a multiplication of two components. The paper conducts theoretical analysis on the proposed framework and also discusses its optimization algorithms.

Submitted by Assigned_Reviewer_44

UPDATE:

After reading the feedback, I now understand that Lemma 1 should be correct. Since the proof also confuses the other reviewer, I suggest the authors to improve the presentation of Lemma 1 to avoid such confusion. Also, may the paper accepted, please emphasize the impact of the equivalence.

===========

This paper studies a particular type of multi-task learning, in which the models are decomposed into the product of two components. The authors have established these class of models with group Lasso models. This idea is really interesting and gives many insights into the two sets of formulations. The authors extend the models based on these insights with new parameters and designed an alternating optimization algorithm to solve the problem. The theoretical contribution of the paper is interesting. However, I have some concerns on this paper.

First of all, the Lemma does not seem to be right. In Lemma1, to satisfy the equality of Cauchy-Schwarz, we need a specific relationship between sigma_j and alpha. However, in (3), we note that both alpha and sigma are variables to be optimized, and in the optimal solution of (3), it is not necessary the case that this relationship holds, and thus the equality of Cauchy-Schwarz may not hold. As such, the causality here does not seem correct to me.

If authors can show that Lemma 1 is correct, then the equivalence of the two types of formulations is novel. Besides the novelty, the authors should further elaborate on why the equivalence is important. The value of such equivalence should be the insights given by it. In this paper, since this equivalence does not have much implication on the efficiency (one is an efficient alternative to the other in some scenario), the new formulations derived based on the equivalence should be the most important part to convince reader its importance. Unfortunately, the paper does not say much about the new formulations, neither does it provide a comprehensive empirical study on them.

Why p and k are chosen to be only 1 or 2? Is there any justification on choices? If there is no theoretical justification, it is also interesting to provide empirical observations on different p, k combinations other than fixing them to be 1 and 2 (e.g., fraction numbers). In the Table 2, we see that in the real datasets, the proposed two formulations MMTFL typically cannot improve much from the existing approaches. This give me concern of how really useful are these newly proposed methods. I would encourage the authors to explore on more real multi-task learning datasets and try to explore what are those datasets that the proposed method works well and why this is the case.
Summary: In the paper, the authors proved the equivalence of two multi-task learning formulations, and derived new formulations based on the equivalence. However, the impact of the finding is limited due to the weak empirical evaluation.
Author Feedback
Author rebuttal: We thank all the reviewers for their constructive feedback.

To Reviewer_17

For lemma 1, we prove if \hat\alpha optimizes (2), it also optimizes (3) together with the specific \sigma. Then we prove if (\hat\alpha,\hat\sigma) is optimal to (3), \hat\alpha also optimizes (2) by proof of contradiction. Please see the response to Reviewer_44 for more explanation.

By the equivalence analysis, the proposed formula corresponds to a family of methods including some convex (if equivalent to convex regularizers), and others non-convex (non-matrix-norm based regularizers) assuming a convex loss function is used. It might be insightful to study the pros and cons of the non-convex regularizers (as discussed in non-convex MTL, e.g.[5]).

Algorithm 1 solves two subproblems in each iteration. Subproblem 1 performs single task learning for each task which can be solved using an existing efficient solver. Subproblem 2 has the closed-form solution, thus requiring a minimal computation cost. Increasing the number of tasks will increase the number of Subproblems 1 to be solved. Hence, the computation cost linearly increases with the number of tasks. It is worth noting that this algorithm was suitable to solve the entire family of MMTFL. More efficient algorithms may be designed for specific choices of MMTFL.

To Reviewer_44

Lemma 1 is correct. The reviewer’s comment suggests us that our proof may be re-structured to be more lucid.

By the Cauchy-Schwarz inequality, the objective function of (2) is the lower bound of the objective function of (3) for any given alpha (including optimal alphas), and the lower bound can be attained if and only if sigma is set to the formula that is a function of alpha.

It is indeed the case that the equality holds with the OPTIMAL solutions of (3); otherwise let (alpha*, sigma*) be an optimal solution to (3) where sigma* and alpha* are not in the relationship. Then we can set a new sigma’ where the jth entry is mu^(1/2) mu^(-1/2) ||alpha*^j||^(p/q)(in the relationship). Then the new (alpha*, sigma’) will bring a lower objective value of (3) than that of (alpha*, sigma*) and reach the lower bound, which contradicts to the optimality of (alpha*, sigma*). Hence, the equality holds for (alpha*, sigma*). Then we prove that this alpha* is optimal to (2). If not, let another alpha’ be optimal to (2), which means the optimal objective of (2) is strictly
less than the objective value of (3) at (alpha*, sigma*). We then use alpha’ to compute a new sigma’ via the relationship. The new (alpha’, sigma’) give a new objective value of (3), which is exactly equal to the optimal objective of (2) but less than that of (3) at (alpha*, sigma*), another contradiction to the optimality of (alpha*,sigma*).

Our proof in the paper, although concise, sketches the main idea. We hope the above paragraph helps to understand this lemma.

On a second note, we believe that proving the equivalence between joint regularized methods and MMTFL is very important, and can bring several research directions. (1) First of all, as pointed out by the reviewer, it helps to better understand the two families of methods, and we now know that they are not different in theory. (2) By proving the equivalence, we have expanded the joint regularized methods that can use non-matrix-norm based regularizers, which has never been considered before, and even considered, the resultant formulations can be difficult to solve. Our MMTFL formula together with Algorithm 1 provides an alternative way to solve them. (3) It is insightful to discuss the data type for which a specific regluarizer is most suitable. We simulated data following the justification in Section 6 for the two new formulas. For instance, simulation 1 was created assuming all tasks share a common but sparse set of features. This is exactly the assumption for the joint regularized methods to work. Even so, the new formula MMTFL(2,1) outperforms other methods (figure 1 left and table 2 top). This is already an insight derived from our equivalence proof that the existing joint regularized methods may not be the best choice for that assumption even though designed for it. We agree that in a longer version of this paper, we will explore more real-world datasets. (4) As pointed out by this reviewer, there are certainly other values of p and k that are worth additional efforts to understand their pros and cons.

To Reviewer_5

We appreciate the advices on the equivalence statement and on the presentation of the paper.

For the concerns with the evaluation, the 3-fold cross validation was performed only on the training data to tune the parameters, and the size of training set was chosen to be 25%, 33% and 50% of all the available examples respectively. The random partition of the training and test sets was repeated 15 times. The regularization parameters were tuned for each partition. For most of the 15 trials, the same value was chosen for the parameter within each choice of MMTFL.

We should have guided the eyeballing. Figure 1 shows the parameter matrices obtained by each of the compared methods. We would like the readers to pay attention to the specific sparsity pattern. The top line shows the true models. In the left figure, a middle area of each bar should be white because those features are not used in the true model. Only MMTFL(2,1) and MMTFL(1,1) show reasonable sparsity over that area, but MMTFL(2,1) shows better weights (darker) on those features that are used than MMTFL(1,1), the common choice. This suggests that the previous method that uses 1-norm on both multiplicative components enforces too strong sparsity, and may not be the best choice. In the right figure, MMTFL(1,1) and MMTFL(1,2) show matrices most similar to the true model, but if we compare the staircase area, MMTFL(1,1) is too sparse on the useful features. Hence, MMTFL(1,2) is the best choice for this type of data among these methods.